# Correlation between the Chemiluminescent Activity of Neutrophilic Granulocytes and the Lipid Peroxidation–Antioxidant Defense System in Gastric Cancer Associated with *Helicobacter pylori* Infection

**DOI:** 10.3390/biomedicines11072043

**Published:** 2023-07-20

**Authors:** Olga Valentinovna Smirnova, Alexander Alexandrovich Sinyakov, Eduard Vilyamovich Kasparov

**Affiliations:** Federal Research Center “Krasnoyarsk Science Center” of the Siberian Branch of the Russian Academy of Sciences, Scientific Research Institute of Medical Problems of the North, St. Partizan Zheleznyaka 3 “G”, 660022 Krasnoyarsk, Russia; sinyakov.alekzandr@mail.ru (A.A.S.); impn@impn.ru (E.V.K.)

**Keywords:** gastric cancer, *H. pylori*, activity of neutrophilic granulocytes, oxidative stress

## Abstract

Aim. To study the processes of lipid peroxidation and the activity of antioxidant defense enzymes depending on the chemiluminescent activity of neutrophilic granulocytes in patients with gastric cancer associated with *H. pylori* infection, depending on the stage. Materials and methods. A total of 39 patients with stage I–II gastric cancer and 30 patients with stage III–IV gastric cancer were examined. A study of the chemiluminescent activity of neutrophilic granulocytes was carried out and the parameters of the lipid peroxidation system and antioxidant protection in plasma were determined using the spectrophotometric method. Statistical data processing was performed using the Statistica 7.0 software package (StatSoft, St Tulsa, OK, USA). Results. In patients with gastric cancer associated with *H. pylori* infection, regardless of stage, the proportion of neutrophilic granulocytes with normal activity did not exceed 1/3 of the total number of patients, and the remaining 2/3 of patients had altered chemiluminescent activity of neutrophilic granulocytes. In patients with gastric cancer, by I–II stage of the disease, the majority revealed a reduced function of neutrophilic granulocytes, and in patients with gastric cancer in stage III–IV of the disease, the majority showed increased chemiluminescent activity of neutrophilic granulocytes. Conclusions. In all patients with gastric cancer associated with *H. pylori* infection, regardless of the stage of the disease, an increase in lipid peroxidation processes with activation of antioxidant defense enzymes was detected. At the same time, there were no statistically significant differences between the indicators of the system lipid peroxidation–antioxidant protection depending on the stage of gastric cancer and the chemiluminescent activity of neutrophilic granulocytes, which likely indicates that all reactive oxygen species produced by neutrophilic granulocytes in the respiratory burst are consumed locally, minimally affecting the development of oxidative stress in the blood plasma.

## 1. Introduction

Among all cancers, gastric cancer is the third most common. Despite numerous studies, in recent years, its prognosis has not improved significantly, and the number of patients who died from this cancer in 2018, according to world epidemiological data, was about 783,000 [1,2,3]. Gastric cancer, like many other types of cancer, has a background of chronic inflammation caused by infection (*H. pylori*) or exposure to environmental factors [4,5].

As precancerous conditions progress, free radical activity increases and the risk of various cancers increases too. Oxidative stress is a source of damage to cell membranes and DNA destruction, followed by metaplasia of epithelial cells of the gastric mucosa into adenocarcinoma [6]. Some scientists come to the conclusion that damage to cell membranes and other structures inside the cell by free oxygen radicals underlies many pathological processes that lead to various diseases. Depending on which structures are damaged—the hereditary substance (*DNA*) or the outer membrane—either an oncological disease develops or other disorders are observed. Reactive oxygen species damage the structure of *DNA*, proteins, and various membrane structures of cells.

The main function of neutrophilic granulocytes is phagocytosis, which allows the elimination of a foreign antigen. The final stage of phagocytosis is associated with the production of reactive oxygen species in the respiratory burst. The greater the number of reactive oxygen species, the more complete the phagocytosis and the higher the functional activity of neutrophilic granulocytes observed. Therefore, the study of the chemiluminescent activity of neutrophilic granulocytes indirectly reflects their functional activity. In addition, there is evidence that neutrophilic granulocytes have a protumor effect in malignant tumor progression in the form of neoangiogenesis activation, tumor cell invasion, and the induction of T-cell immunosuppression [7,8,9]. On the other hand, neutrophils are effector antitumor cells, and the cytotoxic effect of neutrophil granules promotes tumor destruction [10,11]. However, the involvement of the nonspecific link of immunity and the role of neutrophils in carcinogenesis are assessed ambiguously [12,13,14].

Given the relationship between the indicators of the chemiluminescent activity of neutrophilic granulocytes and the lipid peroxidation–antioxidant defense system, we assume that neutrophilic granulocytes contribute to the development of oxidative stress in gastric cancer due to the release of a large amount of reactive oxygen species in the respiratory burst. The greater the functional activity of neutrophilic granulocytes, the more reactive oxygen species are formed, which shift the balance towards prooxidants.

Our purpose is to study the processes of lipid peroxidation and the activity of antioxidant defense enzymes depending on the chemiluminescent activity of neutrophilic granulocytes in patients with gastric cancer associated with *H. pylori* infection, depending on the stage. We plan to study whether the intensity of neutrophil chemiluminescence (the amount of ROS) affects the severity of oxidative stress in gastric cancer.

## 2. Materials and Methods

### 2.1. Subjects

The study included 39 patients with stage I-II gastric cancer and 30 patients with stage III–IV gastric cancer (from 45 to 59 years old, mean age 53 ± 5.7). Clinical examination of patients with gastric cancer was carried out in the “Krasnoyarsk Regional Clinical Oncological Dispensary named after V.I. A.I. Kryzhanovsky” department of oncoabdomial surgery named after N.A. Rykovanov (Figure 1). The diagnosis was made on the basis of clinical, anamnestic, laboratory, and instrumental data by an oncologist. Inclusion of patients in the study and collecting of biological material were carried out upon admission of patients to the hospital before the start of therapy. The study included only primary patients with gastric cancer associated with *Helicobacter pylori* infection. The exclusion criteria were patients on chemotherapy, who had previously received chemotherapy, without infection with *Helicobacter pylori*, with comorbidities, and who refused to participate in the study. All patients with gastric cancer according to the international morphological classification had adenocarcinoma, moderately differentiated according to the international histological classification. The material of the study was venous blood, which was taken from the patients in the morning from 8 to 9 o’clock, on an empty stomach, from the cubital vein, into Vacutainer tubes with separating gel and double clotting activator (silica) and with sodium heparin solution (5 U/mL).

The control group was formed from 100 practically healthy middle-aged blood donors (48.7 ± 3.9 years) without gastroenterological complaints or gastroenterological anamnesis and without changes in the gastric mucosa. The study did not include patients with HIV infection, hepatitis, tuberculosis, gastric ulcer, or concomitant acute and chronic diseases in the acute phase. The study did not include patients who refused to participate in the research study.

The study was conducted with the permission of the Biomedical Ethics Committee at the Federal Research Center “Krasnoyarsk Science Center” of the Siberian Branch of the Russian Academy of Sciences, as well as with the permission of the Ethics Committee of the Krasnoyarsk Regional Clinical Oncological Dispensary named after I.I. A.I. Kryzhanovsky” (protocol No.2 of 2 October 2020). In the work with the examined patients, the ethical principles laid down by the Helsinki Declaration of the World Medical Association were observed. All participants signed an informed consent form to participate in the study.

### 2.2. Endoscopic Examination, Histologic Examination, H. pylori Test, and Gastric Juice Sampling

The study included patients with gastric cancer with verified *Helicobacter pylori* infection, which was proven using histological, urease, and culture methods. All biopsies were examined by an experienced pathologist. To avoid contamination, the endoscope was washed and disinfected through immersion in a detergent solution. Specific antibodies (IgG) to *Helicobacter pylori* were detected using an enzyme immunoassay (BIOHIT HealthCare, Helsinki, Finland). Antibody titers of 30 EIU or more were considered positive, while those less than 30 EIU were considered negative for *H. pylori*. In all patients included in the study, infection of the gastric mucosa with *Helicobacter pylori* was confirmed through histological examination with a modified Giemsa stain, a positive urease test (CLOtest; Delta West, Bentley, Western Australia), a positive culture test for *Helicobacter pylori*, and specific serum antibodies to *Helicobacter pylori*.

### 2.3. Determination of the Chemiluminescent Activity of Neutrophilic Granulocytes

In patients with gastric cancer associated with *Helicobacter pylori* infection, an increase in neutrophilic granulocytes was detected at all stages of the disease, but there were no statistically significant differences to the number from the control group.

ROS are constantly formed in a living cell as products of normal oxygen metabolism. A small amount of ROS play the role of mediators of important intracellular signaling pathways, and increased production of ROS leads to oxidative stress. Primary ROS have a regulatory and moderate antimicrobial effect. The greatest amount of ROS is produced in the process of respiratory burst by neutrophils.

As a method for studying the activity of neutrophilic granulocytes (NG), we used chemiluminescent analysis of spontaneous and induced production of ROS by NG. Assessment of spontaneous and induced chemiluminescence was carried out for 90 min on a BLM-3607 biochemiluminescent analyzer (SKTB Nauka, Krasnoyarsk, Russia). The results were recorded and the analyzer was controlled via a personal computer. The following characteristics were determined: the time for the curve to reach the maximum intensity of chemiluminescence (Tmax is the time of activation of neutrophils), the maximum value of the intensity of chemiluminescence (Imax is the maximum amount of ROS produced by neutrophils per unit time), the area of the chemiluminescence curve (S is the total amount of ROS produced by neutrophils). Luminol (CAS No.: 521-31-3) was used as a chemiluminescence enhancer. Opsonized zymosan (Z4250, Merck (Millipore (Temecula, CA, USA), Sigma-Aldrich (Saint Louis, MO, USA), Supelco) served as the respiratory burst inducer. The enhancement of chemiluminescence induced by opsonized zymosan was evaluated using the ratio of the area of induced (Sind) to the area of spontaneous (Ssp) chemiluminescence and was designated by the activation index. The characteristic and function of neutrophilic granulocytes is enhanced in cancerous tissue. However, given that there is a constant circulation of microphages from blood to tissue, followed by their destruction after phagocytosis, we assume that due to the circulation of tumor antigen in the blood, neutrophils in the blood adapt to these changes; therefore, we can consider them as a necessary experimental model.

### 2.4. “Lipid Peroxidation—Antioxidant Protection” System

The following parameters of the pro-oxidant-antioxidant system (MDA, SOD, CAT, GST, GPO, CP) were studied in the blood serum of the studied patients. To determine lipid peroxidation and the antioxidant system, a spectrophotometric method was used on a Thermo SCIENTIFIC GENESYS 10vis instrument (Thermo Fisher Scientific, Waltham, MA, USA).

#### 2.4.1. Measurement of Malondialdehyde Content (MDA)

MDA is a secondary product of lipid peroxidation. The principle of the method is based on the interaction with 2-thiobarbituric acid (TBA, AppliChem, Darmstadt, Germany) to form a chromogen with an absorption maximum in the red part of the visible spectrum with a wavelength of 532 nm. The measurement of the MDA content is carried out taking into account the molar extinction coefficient of the formed chromogen, equal to 1.56 × 10^5^ M^−1^cm^−1^, and is expressed in µmol/L:$$C=\frac{D532\times V_{p.c.}\times 1000}{V_{np}*\varepsilon *B*d}$$

#### 2.4.2. Measurement of Superoxide Dismutase (SOD) Content

The enzyme superoxide dismutase inhibits the autoxidation reaction of adrenaline (Moscow Endocrine Plant (Moscow, Russia)) in an alkaline environment. The intensity of the adrenaline autoxidation reaction is estimated by an increase in absorption at a wavelength of 347 nm, due to the accumulation of oxidation products and the formation of adrenochrome with an absorption maximum at a wavelength of 480 nm.
$$Unit\;of\;activity\frac{SOD}{гHb}=(\frac{Ex-Eo}{Ex}\;)*\;\frac{100\%*F*V*1000}{50*v*d*C}$$
$$\frac{Ex-Eo}{Ex}\;*\;\frac{100\%}{50}—unit\;of\;activity\;per\;mL\;of\;plasma$$

#### 2.4.3. Measurement of Catalase Content

The determination of catalase activity is based on the formation of a yellow-colored complex of hydrogen peroxide with ammonium molybdate (Russia) that was not destroyed during the catalase reaction. Catalase activity is calculated using the following formula:$$A=\frac{\frac{\Delta A}{c}*V*f}{t*v*d*K*Hb*60}$$

#### 2.4.4. Measurement of Glutathione-S-Transferase Content

The content of the glutathione-S-transferase enzyme is determined by the rate of formation of glutathione-S-conjugates during the interaction of reduced glutathione with 1-chloro-2,4-dinitrobenzene (CDNB, Acros Organics, Geel, Belgium). The increase in the content of conjugates during the reaction is recorded spectrophotometrically at a wavelength of 340 nm (maximum absorption of glutathione-S-CDNB).

The enzyme content is determined using the millimolar extinction coefficient for GS-CDNB at a wavelength of 340 nm, equal to 9.6 mM^−1^ cm^−1^, and determined in micromoles of formed glutathione-S-conjugates per minute per 1 g Hb:$$A=\frac{\Delta E/\min*V_{p.c.}*1000}{\varepsilon *V_{n.}*Hb*d}$$

#### 2.4.5. Measurement of Glutathione Peroxidase Content

Glutathione peroxidase (GPO) catalyzes the reaction between glutathione (GSH) and tert-butyl hydroperoxide (TBH), glutathione S-transferase (GST), and *restored* tert-butyl hydroperoxide (TBH):*GSH* + *TBH* = (*GPO*) = *GST* + *restored*
*TBH*.


Enzyme activity is assessed through the change in the GSH content in samples before and after incubation with a model substrate in the course of a color reaction with dithionitro(bis)benzoic acid (DTNBA, Moscow, Russia). The extinction of the experimental and control samples is measured on a spectrophotometer at a wavelength of 412 nm. It is zeroed in distilled water.

Activity is calculated using the formula
$$A=\frac{\Delta D*V_{p.c.}*1000}{V_{np}*\varepsilon *Hb*d}$$

#### 2.4.6. Measurement of Ceruloplasmin (CP) Content in Blood Serum (Revin Method)

The method is based on the oxidation of n-phenylenediamine (Areolab, Moscow, Russia) with the participation of ceruloplasmin. According to the optical density of the resulting products, the activity of ceruloplasmin is judged.

#### 2.4.7. Measurement of the Coefficient of Oxidative Stress (COS)

An individual ratio of prooxidants to antioxidants was determined for each patient with gastric cancer included in this study (Equation (1)—Formula for calculating COS).

In the formula, *i* are the values of the indicators of the studied patient, and n are the values of the indicators of the control group (proper values). While COS normally tends to 1, with an increase in COS more than 1, a shift in the balance towards prooxidants and the development of oxidative stress are detected.
(1)$$COS=\frac{MDA\;\left(i\right)/MDA(n)}{\left(\frac{SOD\left(i\right)}{SOD(n)}\right)*\left(\frac{CAT\left(i\right)}{CAT\left(n\right)}\right)*\left(\frac{GST\left(i\right)}{GST\left(n\right)}\right)*\left(\frac{GPO\left(i\right)}{GPO\left(n\right)}\right)}$$

## 3. Statistical Analysis

Statistical analysis of the obtained results was performed using the Statistica v.12.0 program (StatSoft Inc., Tulsa, OK, USA). To assess differences in groups, nonparametric Kruskal–Wallis tests (for three or more comparison groups) and Mann–Whitney tests (for pairwise comparison) were used. Comparison of groups on a qualitative binary trait was carried out using a two-sided Fisher’s exact test. Data are presented as median (Me) and interquartile range (Q_25_–Q_75_). To determine the correlations, the Spearman method was used, since at least one of the distributions of the analyzed quantitative characteristics was not normal. When the value of the correlation coefficient |r| ≥ 0.75, the relationship between the signs was assessed as strong, at a coefficient of 0.25 < |r| < 0.75, there was the dependence of the average strength, and at |r| ≤ 0.25, there was a weak degree of correlation. When comparing the signs characterizing the frequency, Fisher’s exact test was used.

## 4. Results

### 4.1. Baseline Patient Characteristics

The mean age of the 169 examined patients and the comparison group was 54.3 years (±9.4). The gastric cancer group included sixty-nine patients and they were randomized after three were excluded due to refusal to sign informed consent. Patients with gastric cancer were divided according to the stages of the disease—patients with stage I–II gastric cancer and patients with stage III–IV gastric cancer. The control and study groups were similar in terms of baseline demographic data (Table 1).

### 4.2. Correlation Relationships between Indicators of Chemiluminescent Activity of Neutrophilic Granulocytes and the System “Lipid Peroxidation—Antioxidant Defence”

When studying the correlation relationships between the indicators of the chemiluminescent activity of neutrophilic granulocytes and lipid peroxidation—antioxidant defense, we found unidirectional relationships in all the studied groups, which made it possible to assume a direct relationship between the functional activity of neutrophils and manifestations of oxidative stress.

In patients with gastric cancer at stages I–II of the disease, five strong correlations were found: Imax spon.—SOD, Imax spon.—GST, Imax spon.—GPO, Imax spon.—CP, Imax induced.—MDA. There were seven average correlation relationships: Imax spon.—MDA, Imax spon.—CAT, Imax induced.—SOD, Imax induced.—CAT, Imax induced.—GST, Imax induced.—GPO, Imax induced.—CP (Table 2).

When analyzing correlations in patients with stage III-IV gastric cancer, we found four strong correlations: Imax spon.—CAT, Imax spon.—GPO, Imax spon.—CP, Imax induced.—SOD and eight average correlation relationships: Imax spon.—MDA, Imax spon.—SOD, Imax spon.—GST, Imax induced.—MDA, Imax induced.—CAT, Imax induced.—GST, Imax induced.—GPO, Imax induced.—CP (Table 3).

### 4.3. Distribution of Groups of Patients with Gastric Cancer Depending on the Activity of Neutrophilic Granulocytes

After the correlation relationships between the indicators of the chemiluminescent activity of neutrophilic granulocytes and the system lipid peroxidation—antioxidant defense were found, we assumed that the greater the functional activity of neutrophilic granulocytes, the more reactive oxygen species are formed, which shift the balance towards prooxidants. In the next stage of the study, we ranked all patients with gastric cancer associated with *H. pylori* infection according to the activity of neutrophilic granulocytes (Table 4).

A total of 69 patients with gastric cancer associated with *Helicobacter pylori* were examined. In patients with stage I–II gastric cancer, the following was found: 10 patients (26% of those examined) showed normal activity of neutrophilic granulocytes (corresponding to the values of the control group); 8 patients (21% of those examined) had increased chemiluminescent (CL) activity of neutrophils; 21 patients (54% of those examined) had reduced activity of neutrophilic granulocytes (*p* < 0.05).

When examining patients at the III–IV stages of the disease, the following was found: 10 patients (33% of those examined) with normal activity of neutrophilic granulocytes; 13 patients (44% of those examined) with increased neutrophil activity (*p* < 0.05); 7 patients (23% examined) with reduced activity of neutrophilic granulocytes. Thus, in the early stages of gastric cancer associated with *H. pylori* infection, patients are dominated by reduced CL activity of neutrophils, and in the later stages, increased activity.

### 4.4. Chemiluminescence of Neutrophilic Granulocytes in Patients with Gastric Cancer at Different Stages of the Disease

In the next stage, we studied the chemiluminescence of neutrophilic granulocytes in patients with stage I–II gastric cancer with low neutrophil activity and stage III–IV gastric cancer with high activity of neutrophilic granulocytes.

It was found that the median values of spontaneous luminescence intensity in patients with stage III–IV gastric cancer increased by 1.5 times compared with the control group and the group of patients with stage I–II gastric cancer (*p*_1–3_ = 0.03; *p*_2–3_ = 0.02) (Table 5). When considering the spontaneous area under the curve, an increase in this indicator was found in patients with gastric cancer at both stages I–II and III–IV compared with the control group (*p*_1–2_ < 0.001, *p*_1–3_ < 0.001). In patients with gastric cancer at stage III–IV, there was an increase in the spontaneous area under the curve compared with patients with gastric cancer at stages I–II (*p*_2–3_ < 0.001). In patients with stage III–IV gastric cancer, an increase in the induced intensity of chemiluminescence of neutrophilic granulocytes was revealed compared with the control group and the group of patients with stage I–II gastric cancer (*p*_1–3_ = 0.04; *p*_2–3_ = 0.01). When studying the induced area under the curve in patients with gastric cancer at stages I–II and III–IV, this indicator increased relative to the control group (*p*_1–2_ < 0.001, *p*_1–3_ < 0.001). In addition, in patients at stages III–IV of the disease, there was an increase in the induced area under the curve compared with patients with gastric cancer at stages I–II (*p*_2–3_ < 0.001).

### 4.5. Indicators of the System “Lipid Peroxidation—Antioxidant Protection” (LPO-AOP) in Gastric Cancer, Depending on the Stage of the Disease

We measured the parameters of the LPO-AOP system in patients with gastric cancer at different stages of the disease. At the same time, in subsequent experiments, data were taken from patients with stage I–II gastric cancer with low activity of neutrophilic granulocytes (21 patients), and in the group with stage III–IV gastric cancer, data were taken from patients in whom the activity of neutrophilic granulocytes was increased (13 people). The content of the LPO product (malonic dialdehyde) and the content of the main enzymes of the antioxidant defense system were studied (superoxide dismutase, catalase, glutathione-S-transferase, glutathione peroxidase, ceruloplasmin).

We found an increase in the median MDA in patients with stage I–II gastric cancer and stage III–IV gastric cancer relative to the control group (MDA: *p*_1–2_ = 0.001; *p*_1–3_ = 0.001;) (Table 6).

Further, the state of the AOP system in the groups of patients was assessed. It was found that the median values of superoxide dismutase, catalase activity, glutathione-S-transferase, glutathione peroxidase, and ceruloplasmin in plasma increased in all groups of patients compared with the control group (SOD: *p*_1–2_ = 0.03; *p*_1–3_ = 0.002, CAT: *p*_1–2_ = 0.03, *p*_1–3_ = 0.03, GST: *p*_1–2_ = 0.04, *p*_1–3_ = 0.046, GPO: *p*_1–2_ = 0.01, *p*_1–3_ = 0.023, CP: *p*_1–2_ < 0.001; *p*_1–3_ < 0.001). In addition, in patients with stage III–IV gastric cancer, there was a significant increase in the level of ceruloplasmin compared with patients with gastric cancer in stages I-II of the disease (CP: *p*_2–3_ < 0.001). The coefficient of oxidative stress reflects the balance of the processes of lipid peroxidation from the antioxidant system and normally tends to 1. When calculating the coefficient of oxidative stress in patients suffering from gastric cancer, it was found that the COS at stages I–II of the disease was 2.8, and at stages III–IV it increased up to 3.9.

## 5. Discussion

When studying the chemiluminescent activity of neutrophilic granulocytes, it was found that in patients with gastric cancer associated with *H. pylori* infection, in the early stages (I–II), the majority of patients showed reduced CL activity of neutrophils, and in the late stages (III–IV), it increased. The revealed difference is probably due to the complex effect of the infectious agent and the tumor factor, while at the early stages, cells may be inhibited, and later they adapt to changes, possibly with the replacement of the function from antitumor to protumor. Neutrophils respond to mediators released by the tumor and its microenvironment, resulting in the activation of either their antitumor or protumor phenotypes. This neutrophil plasticity is due to the influence of transforming growth factor β (TGF-β), β-interferon (IFN-β), and IL-35, as well as the concentration of cytokines and oxygen in the tumor microenvironment [4]. For a long time, the functional diversity of neutrophils was overlooked; however, if the activation of neutrophils causes damage to surrounding tissues, releasing reactive oxygen species and proteolytic enzymes, these cells are also critical for tissue regeneration. This is due to the fact that neutrophils are able to produce growth factors and pro-angiogenic proteins that promote revascularization. Neutrophils can induce macrophage recruitment, which, in turn, supports and accelerates tissue repair [15]. Due to these opposite functions, the role of neutrophils in the development of cancer is ambiguous [16].

N1 neutrophils are characterized by high expression of immunoactivating chemokines and cytokines, including tumor necrosis factor α (TNF-α). TNF-α activates special receptors capable of recognizing a malignant cell and blocking its further division, as well as promoting its necrosis [17]. Intercellular adhesion molecule 1 (ICAM-1), for example, promotes the aggregation and retention of T cells in the tumor niche by binding to the β2-integrin LFA-1 in the melanoma xenograft model, which leads to improved immune control and potentially limits tumor development [18]. Receptor protein Fas contains the intracellular death domain and participates in the programmed mechanism of apoptosis [19]. The antitumor activity of N1 neutrophils is associated with the direct destruction of tumor cells through the production of reactive oxygen species (ROS) and nitric oxide or the induction of apoptosis associated with the activation of Fas/TRAIL. Other mechanisms include antibody-dependent cell-mediated cytotoxicity (ADCC) and activation of T cell function [20].

In other studies, N2 neutrophils have been shown to play a protumor role through several mechanisms. They contribute to tumor development by producing reactive oxygen species (ROS) and reactive nitrogen species (RNS), as well as stimulating tumor cell proliferation by secreting neutrophil elastase (NE), which enhances tumor invasion, metastasis progression [21], and the secretion of matrix metalloproteinase 9 (MMP9), which activates pro-TGF-β and pro-TNF-α growth factors [22]. Neutrophils N2 act as tumor promoters and express the angiogenic chemokines CC and CXC, vascular endothelial growth factor (VEGF), and the CXCR4 receptor, which plays a key role in the penetration of tumor cells through interstitial barriers [23]. Thus, it can be assumed that in our study, the altered CL activity of neutrophils at different stages of gastric cancer is due to their different phenotypes; in gastric cancer, N1 neutrophils are predominant in the early stages, and N2 neutrophils are predominant in the late stages.

At the molecular level, neutrophils, through the release of ROS, damage DNA, which contributes to the emergence and progression of cancer and an increase in the mutation load. On the other hand, neutrophils contribute to the separation of tumor cells from the basement membrane, inhibiting the initial phases of carcinogenesis. The hyperoxic microenvironment limits the accumulation of neutrophils in tumors, but tumor-infiltrating neutrophils show a high antitumor potential due to increased release of ROS, which limit tumor cell proliferation and induce apoptosis [19]. It is possible that such a multidirectional effect of ROS is associated with a different amount of ROS produced by neutrophils. On the one hand, low levels of ROS promote the survival of cancer cells, since the progression of the cell cycle, caused by growth factors and receptor tyrosine kinases (RTKs), requires ROS for activation [24]. On the other hand, a high level of ROS can suppress tumor growth due to the stable activation of the cell cycle inhibitor and induction of cell death [20].

When evaluating the activity of the “lipid peroxidation—antioxidant system” in patients with gastric cancer at stages I–II and III–IV of the disease, we found unidirectional changes in the form of an increase in the amount of malondialdehyde and an increase in the activity of all AOP enzymes, which proves the role of oxidative stress in the development of the tumor process [14,15]. There were no statistically significant differences in LPO-AOP at different stages of gastric cancer, and there was no significant difference in the oxidative stress ratio at different chemiluminescent activity levels of neutrophils. This proves that different amounts of ROS produced by neutrophils are used in different cellular processes.

## 6. Conclusions

For the first time, relationships between the chemiluminescent activity of neutrophilic granulocytes and the compartments of the lipid peroxidation–antioxidant protection system in gastric cancer associated with *H. pylori* infection, depending on the stage, were revealed, which indicates the involvement of microphages in carcinogenesis. It was shown that, regardless of the stage of the disease, correlation relationships were unidirectional.

In patients with gastric cancer associated with *H. pylori* infection, regardless of stage, the proportion of neutrophilic granulocytes with normal activity does not exceed 1/3 of the total number of patients, and the remaining 2/3 of patients have altered chemiluminescent activity of neutrophilic granulocytes. In patients with gastric cancer, by I–II stage of the disease, the majority revealed a reduced function of neutrophilic granulocytes, and in patients with gastric cancer stage in III–IV of the disease, the majority showed increased chemiluminescent activity of neutrophilic granulocytes.

When comparing the altered chemiluminescent activity of neutrophilic granulocytes in gastric cancer, depending on the stage, a twofold increase in Imax spon. and Imax ind. was revealed, and the area under the curve (S spon, S ind) in the spontaneous and induced state of chemiluminescence at stages III–IV of the disease is approximately 3 times that at stages I-II. At the same time, the time to reach the maximum of chemiluminescence at all stages of gastric cancer did not differ from the control, which indicates that cells and neutrophilic granulocytes are activated in the same way.

In all patients with gastric cancer associated with *H. pylori* infection, regardless of the stage of the disease, an increase in lipid peroxidation processes with activation of antioxidant defense enzymes was detected. At the same time, there are no statistically significant differences between the indicators of the system lipid peroxidation—antioxidant protection depending on the stage of gastric cancer and the chemiluminescent activity of neutrophilic granulocytes, which likely indicates that all reactive oxygen species produced by neutrophilic granulocytes in the respiratory burst are consumed locally, minimally affecting the development of oxidative stress in the blood plasma.

In patients with gastric cancer not infected with *Helicobacter pylori*, the chemiluminescent activity of neutrophils increased depending on the stage of the disease, with maximum values at stage IV4 of the disease. This is due to an increase in the amount of tumor antigen, both in the blood and in the tissue of the stomach, and the desire of neutrophils to phagocytize it. When infected with *Helicobacter pylori*, neutrophils encounter a combination of two types of antigens: microbial (foreign species) and tumor (altered own tissue). It seems that this simultaneous combination of antigens is difficult for the rearrangement of neutrophils, hence the change in their chemiluminescent activity, and the possible change in the actual phenotype of neutrophils with the replacement of the function with a protumor one. Therefore, the detection of antibodies to *Helicobacter pylori* is an unfavorable marker for the development and progression of gastric cancer and a predictive biomarker for the aggressiveness of gastric cancer. Chemotherapy and surgical treatment are recommended when *Helicobacter pylori* infection is detected in gastric cancer to eradicate the pathogen.

Limitation of the study: the material of the study was only the blood of patients with only moderately differentiated gastric adenocarcinoma infected with *Helicobacter pylori*.

## Figures and Tables

**Figure 1 biomedicines-11-02043-f001:**
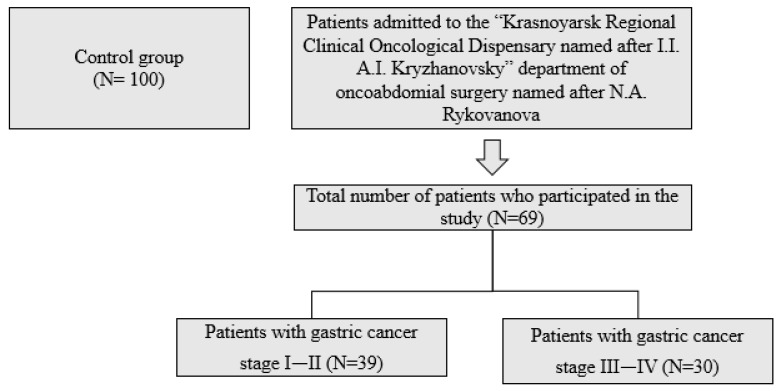
Scheme of inclusion of patients in the study.

**Table 1 biomedicines-11-02043-t001:** Initial demographic data evaluation of two groups.

Parameter	Control Group, N = 100	GC I–II, N = 39	GC III–IV, N = 30	*p*-Value
Gender (n, %)				0.325
Male	58 (58%)	23 (59%)	19 (63%)	
Female	42 (42%)	16 (41%)	11 (37%)	
Age, (y)	47.54 ± 12.65	57.3 ± 9.65	49.87 ± 8.34	0.414
Weight, (kg)	58.4 (±5.3)	56.1 (±3.2)	53.2 (±4.1)	0.256
Height, (m)	1.75 (±0.05)	1.7 (±0.08)	1.69 (±0.05)	0.465
BMI, (kg/m^2^)	22.69 ± 3.71	19.52 ± 3.33	17.22 ± 3.89	0.398
Alcoholic drinks				0.472
1. Never	16 (16%)	9 (23%)	7 (23%)	
2. Past	41 (41%)	16 (41%)	13 (44%)	
3. Current	43 (43%)	14 (36%)	10 (33%)	
Smoking				0.371
1. Never	50 (50%)	12 (31%)	11(37%)	
2. Past	18 (18%)	7 (18%)	11 (37%)	
3. Current	32 (32%)	20 (51%)	8 (26%)	

**Table 2 biomedicines-11-02043-t002:** Correlation relationships in patients with stage I–II gastric cancer.

	MDA	SOD	CAT	GST	GPO	CP
Imax spon.	**r = 0.72** (*p* = 0.03)	**r = 0.78** (*p* = 0.002)	**r = 0.71** (*p* = 0.05)	**r = 0.78** (*p* = 0.01)	**r = 0.78** (*p* = 0.02)	**r = 0.84** (*p* < 0.001)
I max induced	**r = 0.79** (*p* = 0.013)	**r = 0.73** (*p* = 0.021)	**r = 0.74 ** (*p* = 0.012)	**r = 0.73** (*p* = 0.03)	**r = 0.7** (*p* = 0.041)	**r = 0.72** (*p* = 0.004)

**Table 3 biomedicines-11-02043-t003:** Correlation relationships in patients with stage III–IV gastric cancer.

	MDA	SOD	CAT	GST	GPO	CP
Imax spon.	**r = 0.74** (*p* = 0.002)	**r = 0.7** (*p* = 0.002)	**r = 0.75** (*p* = 0.05)	**r = 0.71** (*p* = 0.01)	**r = 0.75** (*p* < 0.001)	**r = 0.8** (*p* = 0.03)
I max induced	**r = 0.7** (*p* = 0.03)	**r = 0.8** (*p* = 0.03)	**r = 0.74** (*p* = 0.004)	**r = 0.73** (*p* = 0.02)	**r = 0.74** (*p* = 0.023)	**r = 0.7** (*p* = 0.02)

**Table 4 biomedicines-11-02043-t004:** Distribution of chemiluminescent activity of neutrophilic granulocytes in patients with gastric cancer at different stages of the disease.

Activity of Neutrophilic Granulocytes	Gastric Cancer I–II N = 39	Gastric Cancer III–IV N = 30
Normal activity of neutrophilic granulocytes (abs., %)	10 patients (26%)	10 patients (33%)
Increased activity of neutrophilic granulocytes (abs., %)	8 patients (21%)	13 patients (44%)
Reduced activity of neutrophilic granulocytes (abs., %)	21 patients (54%)	7 patients (23%)

**Table 5 biomedicines-11-02043-t005:** Chemiluminescent activity of neutrophilic granulocytes in patients with gastric cancer at various stages of the disease, depending on the level of activity of neutrophilic granulocytes (Me [Q_25_–Q_75_]).

Indicators	Control Group (N = 100), (1)	GC I–II, Reduced Activity of Neutrophilic Granulocytes (N = 21), (2)	GC III–IV, Increased Activity of Neutrophilic Granulocytes (N = 13), (3)
Imax spontaneous (y.e.)	**10,730** (3262–21,997)	**8215** (1239–11,473)	**25,285** (24,472–26,787)
			*p*_1–3_ = 0.03; *p*_2–3_ = 0.02
Squr spontaneous (×10^6^)	**0.22** (0.15–0.4)	**0.11** (0.09–0.7)	**9.6** (7.2–9.9)
		*p*_1–2_ < 0.001	*p*_1–3_ < 0.001; *p*_2–3_ < 0.001
T max spont. (s.)	**969** (615–1753)	**1102** (801–1436)	**1278** (1187–2116)

Imax induced (y.e.)	**19,904** (7281.5–32,121)	**17,625** (6525–30,530)	**33,135.5** (32,635.5–63,446.5)
			*p*_1–3_ = 0.04; *p*_2–3_ = 0.01
Squr induced (×10^6^)	**0.4** (0.15–0.6)	**0.34** (4.9–6.1)	**12.8** (11.7–13.5)
		*p*_1–2_ < 0.001	*p*_1–3_ < 0.001; *p*_2–3_ < 0.001
T max induced (s.)	**1380.8** (796–1586)	**1204** (700–1525)	**1456** (1105–1505)
Activation index	**1.3** (0.9–2.0)	**1.3** (1.01–2.02)	**2.1** (1.8–2.7)
			*p*_1–3_ = 0.001

Note: *p*_1–2_: statistically significant differences between the group of patients with stage I–II gastric cancer and the control group. *p*_1–3_ statistically significant differences between the group of patients with stage III–IV gastric cancer and the control group, *p*_2–3_: statistically significant differences between the group of patients with stage I–II gastric cancer and the group of patients with stage III–IV gastric cancer.

**Table 6 biomedicines-11-02043-t006:** LPO-AOP parameters in patients with gastric cancer at various stages of the disease, depending on the level of activity neutrophilic granulocytes (Me [Q_25_–Q_75_]).

Indicators	Control Group (N = 100), (1)	GC I–II, Reduced Activity of Neutrophilic Granulocytes (N = 21), (2)	GC III–IV, Increased Activity of Neutrophilic Granulocytes (N = 13), (3)
MDA, µmol/1 g of protein	**1.6** (0.96–2.24)	**47.83** (36.7–81.1)	**55.3** (50.3–66.75)
		*p*_1–2_ = 0.001	*p*_1–3_ = 0.001
SOD, U/min/1 g protein	**204.41** (151.05–250.3)	**267.1** (185.8–421.6)	**367.2** (272.2–431.3)
		*p*_1–2_ = 0.03	*p*_1–3_ = 0.002
CAT, µmol/s/1 g of protein	**0.27** (0.16–0.39)	**0.77** (0.61–0.88)	**0.62** (0.5–0.8)
		*p*_1–2_ = 0.03	*p*_1–3_ = 0.03
GST, mmol/min/1 g protein	**41.3** (37.7–42.64)	**76.3** (52.1–93.6)	**64.2** (55.6–83.2)
		*p*_1–2_ = 0.04	*p*_1–3_ = 0.046
GPO, µmol/1 g protein	**107.9** (81.19–162.38)	**174.7** (147.3–200.1)	**150.8** (134.7–156.2)
		*p*_1–2_ = 0.01	*p*_1–3_ = 0.023
CP, mg/L	**192.5** (157.5–227.5)	**481.5** (321.5–729.1)	**767.1** (649.5–806.8)
		*p*_1–2_ < 0.001	*p*_1–3_ < 0.001; *p*_2–3_ < 0.001

Note: *p*_1–2_–statistically significant differences between the group of patients with stage I–II gastric cancer and the control group. *p*_1–3_–statistically significant differences between the group of patients with stage III–IV gastric cancer and the control group, *p*_2–3—_statistically significant differences between the group of patients with stage I–II gastric cancer and the group of patients with stage III–IV gastric cancer.

## Data Availability

All data are published in this article, and can be used by other scientists.

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
