# Peer review of "Correlation between the Chemiluminescent Activity of Neutrophilic Granulocytes and the Lipid Peroxidation–Antioxidant Defense System in Gastric Cancer Associated with *Helicobacter pylori* Infection"

_biomedicines, 2023, doi:10.3390/biomedicines11072043_

Round 1
Reviewer 1 Report
1. Chemotherapy has been reported to increase ROS, are the patients recruited for this study never received chemotherapy?
2. How similar are the functions and characteristics of neutrophilic granulocytes in the blood and gastric cancer?
3. What is the amount of neutrophil granulocytes in the blood? And besides neutrophils granulocytes, is there nothing that affects the confirmed increase in ROS?
4. In table 4, compared to control group, 54 percent of patients in the early stages had reduced activity of neutrophilic granulocytes, so why are all indicators higher than the control group in tables 5 and 6?
5. This study was conducted in patients with gastric cancer associated with H. pylori infection, so what is the difference between non-infected and infected patients? And is H. pylori infection involved in the differences in CL activity seen between early and late stages? If so, how is H. pylori infection likely to affect it?

Minor editing of English language required
Author Response
REPLY TO THE REVIEWER.
Dear reviewer, hello!
Thank you very much for your review of our article. I answer your questions:
- Chemotherapy has been reported to increase ROS, are the patients recruited for this study never received chemotherapy?
The study included only primary patients with gastric cancer associated with Helicobacter pylori infection. The exclusion criteria were patients on chemotherapy who had previously received chemotherapy, without infection with Helicobacter pylori, with comorbidities and who refused to participate in the study.
- How similar are the functions and characteristics of neutrophilic granulocytes in the blood and gastric cancer?
The characteristic and function of neutrophilic granulocytes is enhanced in cancerous tissue, however, given that there is a constant circulation of microphages from blood to tissue, followed by their destruction after phagocytosis, we assume that due to the circulation of tumor antigen in the blood, neutrophils in the blood adapt to these changes, therefore, we can consider them as a necessary experimental model.
- What is the amount of neutrophil granulocytes in the blood? And besides neutrophils granulocytes, is there nothing that affects the confirmed increase in ROS?
In patients with gastric cancer associated with Helicobacter pylori infection, an increase in neutrophilic granulocytes was detected at all stages of the disease, but there were no statistically significant differences in the number from the control group.
ROS are constantly formed in a living cell as products of normal oxygen metabolism. A small amount of ROS plays the role of mediators of important intracellular signaling pathways, increased production of ROS leads to oxidative stress. Primary ROS have a regulatory and moderate antimicrobial effect. The greatest amount of ROS is produced in the process of respiratory burst by neutrophils.
- In table 4, compared to control group, 54 percent of patients in the early stages had reduced activity of neutrophilic granulocytes, so why are all indicators higher than the control group in tables 5 and 6?
Indicators of reduced chemiluminescent activity of neutrophils in early-stage gastric cancer in Table 5 have been corrected (there was an error). In table 6, all indicators are correct, and with reduced and increased neutrophil function in gastric cancer, depending on the stage, there is an increase in the indicators of the lipid peroxidation system and antioxidant protection relative to the control group.
- This study was conducted in patients with gastric cancer associated with H. pylori infection, so what is the difference between non-infected and infected patients? And is H. pylori infection involved in the differences in CL activity seen between early and late stages? If so, how is H. pylori infection likely to affect it?
In patients with gastric cancer not infected with Helicobacter pylori infection, the chemiluminescent activity of neutrophils increased depending on the stage of the disease, with maximum values at stage 4 of the disease. This is due to an increase in the amount of tumor antigen, both in the blood and in the tissue of the stomach and the desire of neutrophils to phagocytize it. When infected with Helicobacter pylori, neutrophils encounter a combination of two types of antigens: microbial (foreign species) and tumor (altered own tissue). Apparently, this simultaneous combination of antigens is difficult for the rearrangement of neutrophils, hence the change in their chemiluminescent activity, and the possible change in the actual phenotype of neutrophils with the replacement of the function with a protumor one. Therefore, the detection of antibodies to Helicobacter pylori is an unfavorable marker for the development and progression of gastric cancer. It is recommended that when Helicobacter pylori infection is detected in gastric cancer, along with chemotherapy and surgical treatment, eradicate the pathogen.
All changes have been made to the article and are highlighted in green.

Reviewer 2 Report
1. In title word "relationship" could be replaced by correlation or asocciation.
2. Abstract is well prepared. H.pylori in few sentences is not written by italic (also in other parts of the manuscript)
3. In the Introduction, sentence belowe should be modified:
"The purpose of this study: to study the processes of lipid peroxidation and the activity of antioxidant defense enzymes depending on the chemiluminescent activity of neutrophilic granulocytes in patients with gastric cancer associated with H. pylori infection, depending on the stage."
4. The number of investigated groups (30+39 after division) is low when compare to the control group. Please add table with detailed clinicopathological characterisation of investigated groups. Histological grade and others parameters also can influence on your results. For clinicopathological parameters also statistical analysis should be done.
5. The reagents names should be added, concentration and steps of analysis. There is only information about instrument, description of the mathematical calculation and the principle of the method.
6.There are fragments in the article marked in green, written in different fonts. Everything should be corrected.
7. The paragraph conclusion is too long, many elements should be moved to discussion. Last paragraph marked in this version in green is not conclusion from this study. The authors should write conclusion on the basis of obtained results and point clinical potential of th study.
8.The limitations of the study should be pointed (like diversity of the investigated group, the low number of cases). The main limitation is that material for study is only blood not tissue.
Author Response
REPLY TO THE REVIEWER!
Dear reviewer, hello!
In response to your comments and questions:
- In title word "relationship" could be replaced by correlation or association.
In the title, the word "relationship" was replaced by the word "correlation".
- Abstract is well prepared. H. pyloriin few sentences is not written by italic (also in other parts of the manuscript).
In the abstract and throughout the article, H. pylori is in italics.
- In the Introduction, sentence below should be modified:
"The purpose of this study: to study the processes of lipid peroxidation and the activity of antioxidant defense enzymes depending on the chemiluminescent activity of neutrophilic granulocytes in patients with gastric cancer associated with H. pylori infection, depending on the stage."
The sentence was changed to the following.
The purpose is to study the processes of lipid peroxidation and the activity of antioxidant defense enzymes depending on the chemiluminescent activity of neutrophilic granulocytes in patients with gastric cancer associated with H. pylori infection, depending on the stage.
- The number of investigated groups (30+39 after division) is low when compare to the control group. Please add table with detailed clinicopathological characterisation of investigated groups. Histological grade and others parameters also can influence on your results. For clinicopathological parameters also statistical analysis should be done.
The study included 69 primary patients infected with Helicobacter pylori infection. The exclusion criteria were patients on chemotherapy who had previously received chemotherapy, without infection with Helicobacter pylori, with comorbidities and who refused to participate in the study. This explains the small number of patients included in the study. All patients with gastric cancer according to the International morphological classification had adenocarcinoma, moderately differentiated according to the International histological classification. Given that there was a careful personalized selection of patients in the study, all patients with moderately differentiated gastric adenocarcinoma were included in the study, there is no need for additional statistical analysis on the clinical and pathological parameters of the tumor itself. Patients are divided into stages.
- The reagents names should be added, concentration and steps of analysis. There is only information about instrument, description of the mathematical calculation and the principle of the method.
Information added.
- There are fragments in the article marked in green, written in different fonts. Everything should be corrected.
Everything is fixed.
7.The paragraph conclusion is too long, many elements should be moved to discussion. Last paragraph marked in this version in green is not conclusion from this study. The authors should write conclusion on the basis of obtained results and point clinical potential of the study.
The conclusion includes the main results of the study. The last paragraph is our proposal for the clinical application of the obtained results (taking into account the wishes of other reviewers).
8.The limitations of the study should be pointed (like diversity of the investigated group, the low number of cases). The main limitation is that material for study is only blood not tissue.
Research limit added.
Limitation of the study: the material of the study was only the blood of patients with only moderately differentiated gastric adenocarcinoma infected with Helicobacter pylori.
All changes have been made to the article and are highlighted in green.

Reviewer 3 Report
The authors in this manuscript "Relationship between the chemiluminescent activity of neutrophil granulocytes and the antioxidant-lipid peroxidation defense system in gastric cancer associated with Helicobacter pylori infection" correlate the lipid peroxidation processes and the activity of antioxidant defense enzymes depending on the chemiluminescent activity of neutrophil granulocytes with gastric cancer associated with H. pylori infection, depending on the stage. This study is conducted on a total of 69 cancer patients, a greater number of patients is necessary to have more reliable data
The authors point out that "this is the first time that the relationships between the chemiluminescent activity of neutrophil granulocytes and the lipid peroxidation-antioxidant protective system compartments have been revealed in gastric cancer associated with H. pylori infection, depending on the stage, indicating the involvement of microphages in carcinogenesis,” but the authors do not explain the application of these results. That is, can the assays they apply be used as predictive biomarkers for the aggressiveness of gastric cancer? o in patients with Helicobacter pylor infection can the development of gastric cancer be predicted on the basis of these values? The authors must describe the purpose of their research, otherwise it seems like a simple manuscript describing a phenomenon, without explaining why.
Author Response
REPLY TO THE REVIEWER.
Dear reviewer, hello!
Thank you very much for your review of our article. I answer your questions:
- The authors in this manuscript "Relationship between the chemiluminescent activity of neutrophil granulocytes and the antioxidant-lipid peroxidation defense system in gastric cancer associated with Helicobacter pylori infection" correlate the lipid peroxidation processes and the activity of antioxidant defense enzymes depending on the chemiluminescent activity of neutrophil granulocytes with gastric cancer associated with H. pylori infection, depending on the stage. This study is conducted on a total of 69 cancer patients, a greater number of patients is necessary to have more reliable data.
The study included 69 primary patients infected with Helicobacter pylori infection. The exclusion criteria were patients on chemotherapy who had previously received chemotherapy, without infection with Helicobacter pylori, with comorbidities and who refused to participate in the study. This explains the small number of patients included in the study.
- The authors point out that "this is the first time that the relationships between the chemiluminescent activity of neutrophil granulocytes and the lipid peroxidation-antioxidant protective system compartments have been revealed in gastric cancer associated with H. pylori infection, depending on the stage, indicating the involvement of microphages in carcinogenesis,” but the authors do not explain the application of these results. That is, can the assays they apply be used as predictive biomarkers for the aggressiveness of gastric cancer? o in patients with Helicobacter pylori infection can the development of gastric cancer be predicted on the basis of these values? The authors must describe the purpose of their research, otherwise it seems like a simple manuscript describing a phenomenon, without explaining why.
In patients with gastric cancer not infected with Helicobacter pylori infection, the chemiluminescent activity of neutrophils increased depending on the stage of the disease, with maximum values at stage 4 of the disease. This is due to an increase in the amount of tumor antigen, both in the blood and in the tissue of the stomach and the desire of neutrophils to phagocytize it. When infected with Helicobacter pylori, neutrophils encounter a combination of two types of antigens: microbial (foreign species) and tumor (altered own tissue). Apparently, this simultaneous combination of antigens is difficult for the rearrangement of neutrophils, hence the change in their chemiluminescent activity, and the possible change in the actual phenotype of neutrophils with the replacement of the function with a protumor one. Therefore, the detection of antibodies to Helicobacter pylori is an unfavorable marker for the development and progression of gastric cancer, predictive biomarker for the aggressiveness of gastric cancer. It is recommended that when Helicobacter pylori infection is detected in gastric cancer, along with chemotherapy and surgical treatment, eradicate the pathogen.
All changes have been made to the article and are highlighted in green.

Round 2
Reviewer 2 Report
The publication is corrected according suggestions.
Reviewer 3 Report
Thank you for the exhaustive answers and additions made to the manuscript.
Author Response
REPLY TO THE REVIEWER.
Dear reviewer, hello!
Thank you very much for your work. Thanks a lot.
